# Better sturdy or slender? Eurasian otter skull plasticity in response to feeding ecology

**Luca Francesco Russo**[1], **Carlo Meloro**[2]*, **Mara De Silvestri**[1], **Elizabeth A. Chadwick**[3], **Anna Loy**[1]

**1** EnvixLab, Department of Biosciences and Territory, Università degli Studi del Molise, Pesche, Italy,
**2** Research Centre in Evolutionary Anthropology and Palaeoecology, Liverpool John Moores University, Liverpool, United Kingdom, **3** Cardiff University, Biomedical Science Building, Museum Avenue, Cardiff, United Kingdom

* C.Meloro@ljmu.ac.uk

**Data Availability Statement:** All relevant data are within the paper and its Supporting Information files

## Abstract

Otters are semi-aquatic mammals specialized in feeding on aquatic prey. The Eurasian otter *Lutra lutra* is the most widely distributed otter species. Despite a low degree of genetic variation across its European range, the population from Great Britain exhibits distinct genetic structuring. We examined 43 skulls of adult Eurasian otters belonging to 18 sampling localities and three genetic clusters (Shetlands, Wales and Scotland). For each sample location, information regarding climate was described using bioclimatic variables from WorldClim, and information on otter diet was extracted from the literature. By using photogrammetry, 3D models were obtained for each skull. To explore any evidence of adaptive divergence within these areas we used a three dimensional geometric morphometric approach to test differences in skull size and shape between areas with genetically distinct populations, as well as the influence of diet, isolation by distance and climate. Males were significantly larger in skull size than females across all the three genetic clusters. Skull shape, but not size, appeared to differ significantly among genetic clusters, with otters from Shetland exhibiting wider zygomatic arches and longer snouts compared to otters from Wales, whereas otters from Scotland displayed intermediate traits. A significant relationship could also be found between skull shape variation, diet as well as climate. Specifically, otters feeding on freshwater fish had more slender and short-snouted skulls compared to otters feeding mostly on marine fish. Individuals living along the coast are characterised by a mixed feeding regime based on marine fish and crustaceans and their skull showed an intermediate shape. Coastal and island otters also had larger orbits and eyes more oriented toward the ground, a larger nasal cavity, and a larger distance between postorbital processes and zygomatic arch. These functional traits could also represent an adaptation to favour the duration and depth of diving, while the slender skull of freshwater feeding otters could improve the hydrodynamics.

**Funding:** The author(s) received no specific funding for this work.

**Competing interests:** NO authors have competing interests.

## Introduction

Otters are semi-aquatic mammals found in a wide variety of aquatic environments, including rivers, lakes, reservoirs, swamps, and marshes [1]. The Eurasian otter *Lutra lutra* is the most widely distributed species within the subfamily Lutrinae, with 11 subspecies occurring from Europe throughout Asia and North Africa [2, 3]. The European populations belong to the nominal subspecies *L. l. lutra* and are currently recovering after a strong decline in the last century due to direct persecution, hunting for pelts, water pollution, and habitat degradation [2]. Wide-ranging mammals often show differences in both shape and size related to ecological gradients [4–6], which are more likely to occur in highly specialized species such as the semi-aquatic otters [7]. Research at range-wide scales has revealed only a low degree of phenotypic and genetic variation across Europe (e.g. [8, 9]), but smaller scale studies have shown evidence both for cranial differentiation (e.g. in Denmark, Germany and East Asia [10–12]), and genetic sub-structuring (e.g. in UK [13, 14], and southern Italy [15]). The skull is a complex anatomical adaptive structure, enclosing the central nervous system and the specialized sense organs and hosting the teeth used to capture, kill the prey, process and manipulate food [16]. Several studies have shown that in mammals, skull shape is closely related to diet [17–21], and shape changes in response to feeding habits can occur among different populations of the same species [22]. The Eurasian otter is an opportunistic feeder able to exploit different aquatic prey depending on their availability and catchability [23]. Its diet includes mainly fish, but also crustaceans, amphibians, reptiles, and, to a lesser extent, birds and mammals [23–27]. This high feeding plasticity is expected to be reflected in the cranial and mandibular morphology with modifications expected, especially in the dentition and masticatory muscle attachment area, as observed for members of the order Carnivora [21, 28]. Furthermore, differences in diet may also be reflected in sexual dimorphism of the skull with respect to size and shape [29–32], as specific adaptations can improve the fitness and reduce intraspecific resource competition [33]. Although mustelids are known usually to show sexual dimorphism only with respect to size, and not shape [7, 29–31], advances in methodology may now permit more nuanced analyses. The advent of geometric morphometric (GM) approaches offer a powerful tool to investigate the role of geographical and ecological gradients in influencing the size and the shape of biological structures, especially the highly informative skull, as well as to explore the role of sexual dimorphism and allometry [34–37]. The study of geographic variation is particularly interesting when dealing with islands, as isolation from the mainland may accelerate the emergence of adaptive traits in highly specialized feeders like the Eurasian otter [38]. Here, we investigate the morphological variation of otters across the mainland and the islands of Great Britain to explore the ultimate drivers of the observed patterns in terms of the genetic, latitudinal and ecological differentiation revealed by recent studies. Otter populations in Britain are ecologically heterogenous, with high levels of genetic sub-structuring (e.g. [13, 14]), differences in scent gland secretions between genetically distinct regions ('odour dialects' [39]), and regional variation in diet [26]. Farnell et al. [40] gave a first insight into 3D morphological variation of skulls across Great Britain, suggesting that the observed changes in size and shape might reflect genetic differences among populations. However, the authors suggested that regional differences could be driven by other potentially confounding factors and that more rigorous research was needed.

We specifically used a 3D GM approach to examine the size and shape variation of Eurasian otter skulls in order to: *i)* evaluate the occurrence of sexual size (SSD) and sexual shape (SShD) dimorphism; *ii)* detect any pattern in the size and shape of skull that could reflect genetic clustering; *iii)* evaluate the role of diet and isolation by distance on size and shape variation; iv) evaluate climate has a proxy of diet adaptation as reported in many studies.

## Methods

### Data collection

We examined 43 skulls of adult Eurasian otters stored in the National Museums of Scotland. Specimens belong to 18 sampling localities (S1 and S2 Tables). Each sample locality was assigned to inland, coastal, or island waters, and pooled in three likely genetic clusters based on Hobbs et al. [14] and Stanton et al. [13]. Stanton et al. [13] describe a distinct contrast in genetic structure between 'Northern Britain' and 'Southern Britain', but do not include samples from the Shetland Isles, which were previously shown distinct from the Scottish mainland by Hobbs et al. [14]. In the current study we used samples originating from the Shetland Islands (F = 8, M = 7, hereafter referred to as 'Shetland'), mainland Scotland, the outer Hebrides, and the Orkney Islands (hereafter referred to as 'Scotland' F = 6, M = 12) and from Wales and central England (hereafter referred to as 'Wales', F = 3; M = 7) (Fig 1).

Genetically, we assume that the 'Wales' samples are part of the Southern Britain population defined by Stanton et al. [13] while the 'Scotland' samples are assumed to be part of the 'Northern Britain' population, although note that samples from the outer Hebrides were not included in either Hobbs et al. [14], or Stanton et al. [13] and may be distinct.

Each skull was placed on a turntable and photographed every 10° on the dorsal, ventral, and vertical projections, for a total of 108 pictures. All pictures were taken using a Canon 30EOS SLR with a fixed 50mm lens, placed on the tripod at a fixed distance (50 cm) from the turntable and activated with a remote control to avoid blurring [41].

Three dimensional pictures were reconstructed using the photogrammetry method [42] by means of Agisoft PhotoScan software (Agisoft PhotoScan, http://www.agisoft.ru/). The resulting models were scaled to real size using Tpsdig [43] and Meshlab [44]. Previous studies have abundantly demonstrated that the level of accuracy generated by 3D photogrammetry models in quantifying size and shape using GM is as high as the one provided by CT scan or laser scanner [45, 46], and these techniques can be implemented within the same analysis without generating significant intra-individual error in GM data [41, 47]. On each 3D model, we positioned thirty 3D landmarks (LM, S1 File) on clearly distinguishable, homologous, and significant anatomical regions, using the software Meshlab (Fig 2).

### Morphometric analyses

A Generalized Procrustes Alignment (GPA) analysis was run to translate, scale and rotate original coordinates [34], using the *gpagen()* function implemented in the R Geomorph package [48, 49]. We used the log transformed centroid size (CS hereafter), i.e. the square root of the sum of squared distances of each landmark, as a proxy of skull size [34]. To avoid noise due to either directional or fluctuating asymmetry, shape changes were explored retaining only the symmetric component of skull shape [50]. The symmetric component was extracted using geomorph's *bilat.symmetry()* function.

A Principal Components Analysis (PCA) was run on the aligned coordinates (i.e. shape variables) of the symmetric component to explore shape changes among genetic clusters and sexes, using the geomorph's *gm.prcomp()* function.

ANOVA on CS and Procrustes ANOVA on shape variables were run to test for the effect of sex, genetic cluster, and their interaction, on size and shape respectively. Procrustes ANOVA was also used to test for static allometry (sensu Klingenberg, [37]), by considering the effect of size (lnCS) and genetic clusters, and their interaction on shape. All analyses were run through the function *procd.lm()*. To analyse the relationship between either size or shape, we used the mean size and the mean shape for each sample locality. Specifically, the mean shape was

computed through the geomorph *mshape*() function as recommended in previous ecogeographical studies [4, 51–53].

## Shape and diet

We explored the existing literature to gather information on otter diet at each sample location. When information was not available, we used references from the closest areas within the

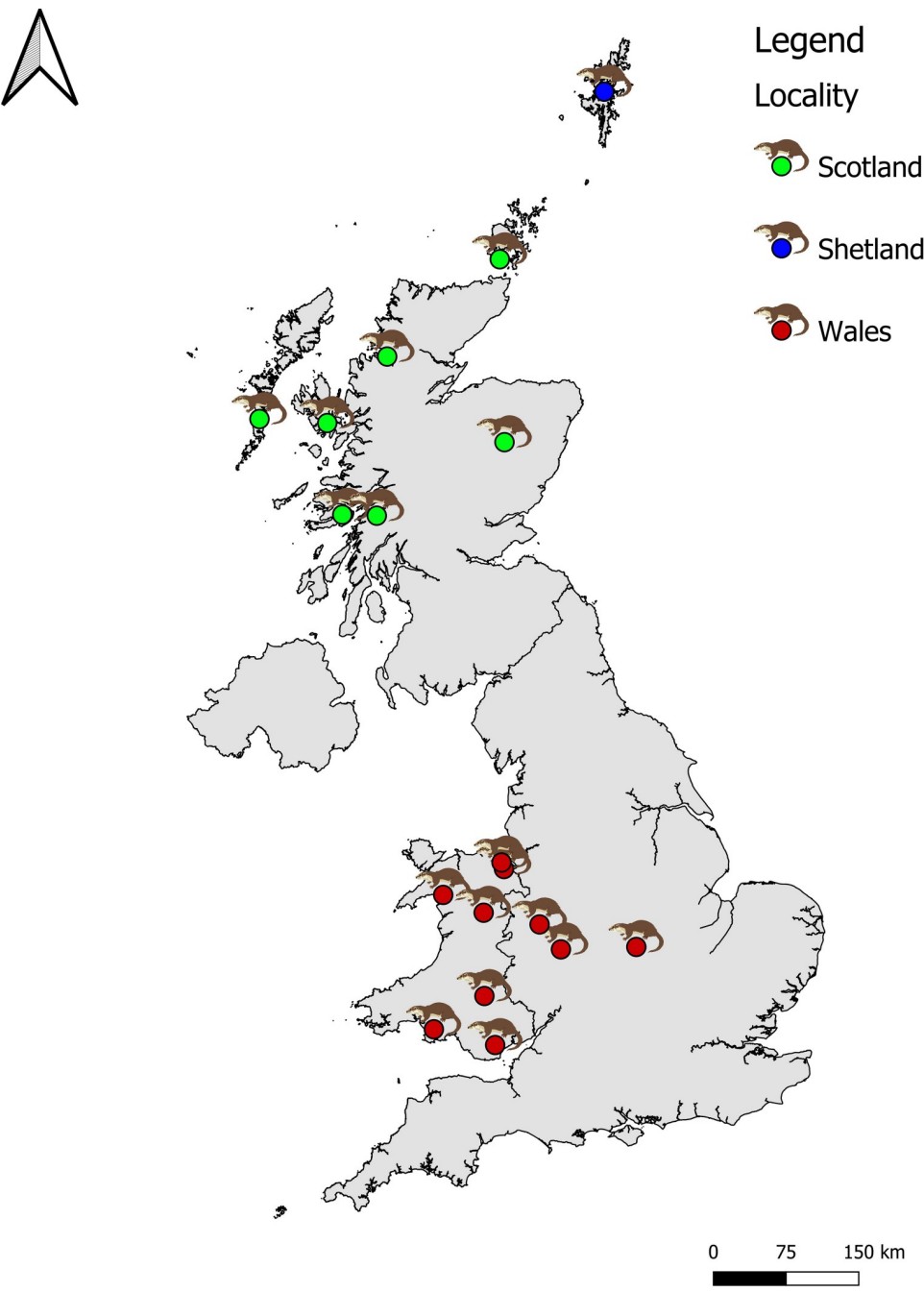

**Fig 1. Geographic origin of otter skulls.** Coloured points represent sampling locations. Sample size for each locality is reported in S1 Table. Country Boundaries was downloaded from www.data.gov.uk under the Open Government Licence v3.0.

same river basin (S3 Table). We specifically recorded the frequencies of occurrence of seven prey categories: marine fish, freshwater fish, crustaceans, amphibians, birds, mammals, plants, and insects (S3 Table). A PCA was then run on the resulting data matrix to explore the variation in the diet of otters living in inland freshwater, mainland coasts, and islands.

We further explored the diet composition by quantifying the frequency of 31 fish families (S3 Table).

The correlation between one or the other diet matrix (i.e. the whole prey categories and the fish families) and either the skull size or shape was explored through a Partial Least Squares regression (PLS, [54]) using *two.b.pls*() function in geomorph, an approach successfully used in previous studies on other mammals and bird species [18, 55, 56].

To explore if the climate can be used as proxy of diet in the Eurasian otter, the results of PLS on climate and PLS on diet were compared using the function *compare.pls ()* of geomorph to assess possible difference or parallelism in shape co-variation with dietary or climatic factors [57].

### Shape and climate

We extracted 19 bioclimatic variables at each sample locality (S4 Table). Variables were recorded at 5min (~ 10 km$^2$) resolution from WorldClim [58]. We used the *vif()* function of the usdm package [59] to account for autocorrelation. The final six uncorrelated variables were standardised for further analyses. To explore climatic pattern across the sample localities and avoid bias due to spatial autocorrelation [60] we used the function Principal Coordinates of Neighbor Matrices (PCNM) [60] implemented in the *pcnm()* function of the vegan package [61] to extract spatial vectors based on specimen locations. Partial Least Squares regression (PLS) was then run between cranial size or shape of each sample locality vs a matrix including the first PCNM scores and the selected bioclimatic variables [62], using geomorph function *two.b.pls()*.

All Procrustes ANOVA and PLS analyses were run using randomized residuals permutation procedures (RRPP) with 1000 permutations.

## Results

### Sexual dimorphism

ANOVA showed a significant difference in size between males and females (Table 1), with male skulls always being larger than females in each of the three genetic clusters (Fig 3). These differences were consistent among genetic clusters since the interaction term "sex: genetic clusters" was not significant.

In contrast, no sexual dimorphism in skull shape could be identified by Procrustes ANOVA, nor in the sex:genetic clusters interaction (Table 1). These results allowed pooling of sexes and including the undetermined specimens for subsequent analyses of shape variation.

### Geographic variation

Otters belonging to the three genetic clusters i.e. Wales, Scotland, and Shetland, showed significant differences in their cranial morphology, accounting for ca. 16% of skull shape variation (Table 1). The Principal Components Analysis (PCA) on shape variables showed a clear separation of genetic clusters along PC1 (16.88% of variance) (Fig 4). 3D wireframe plots related to variation along PC1 indicate that the most northern cluster (otters from Shetland) exhibited a squatter skull with wider zygomatic arches and longer snout, compared to the most southern cluster (from Wales), whereas otters from Scotland displayed an intermediate shape between the two former clusters (Fig 4).

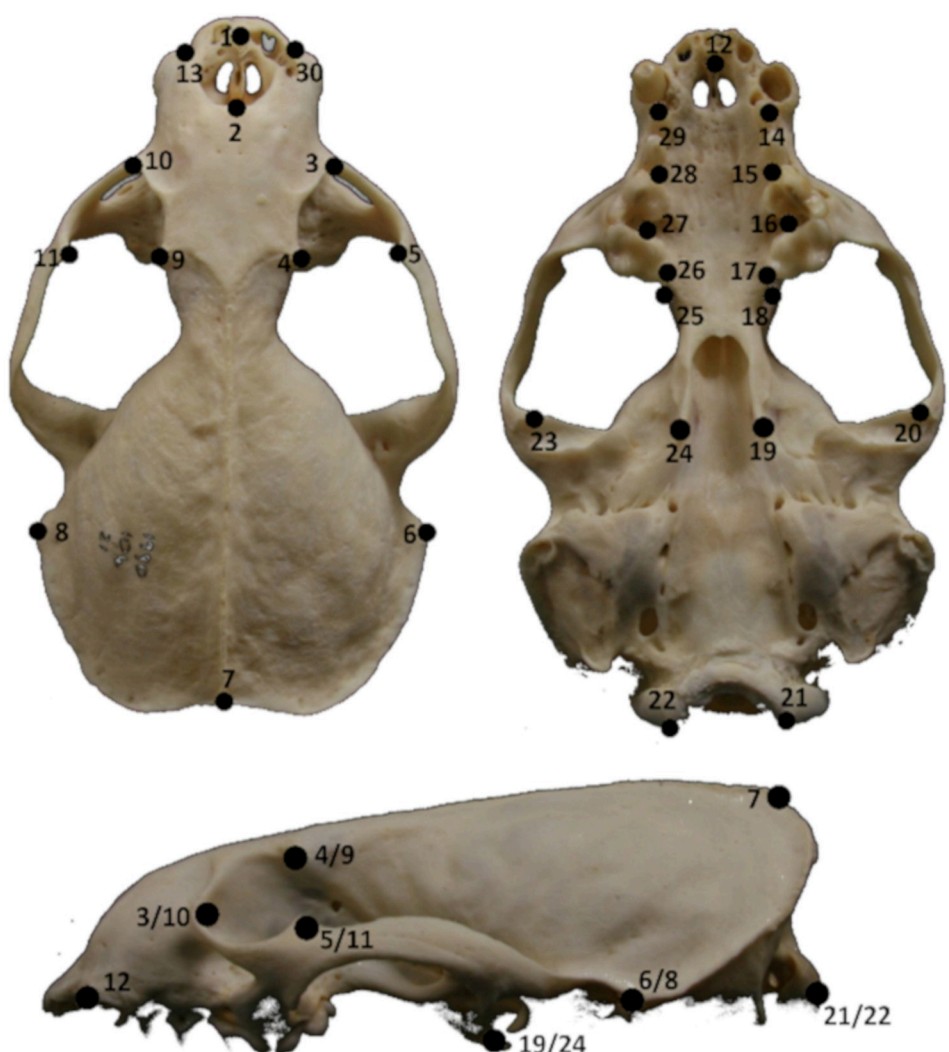

**Fig 2. Location of anatomical landmarks collected on 3D models of otter skulls.** Landmarks are defined as follows:
1 = Superior premaxilla; 2 = nasal suture; 3–10 = lacrimal; 4–9 = postorbital process 5–11; = zygomatic arch;6–
8 = mastoid process; 7 = intersection of temporal line and sagittal crest; 12 = prosthion; 13–30 = premaxilla and nasal
bone suture; 14–29 = canine alveoli; 15–28 = carnassial alveoli; 16–27 = carnassial alveoli; 17–26 = toothrow; 18–
25 = palatine; 19–24 = pterygoid; 20–23 = glenoid cavity; 21–22 = occipital condyle.

These shape differences were not due to allometric shape changes (Table 2) since variation
explained by size was minimal and non-significant when compared to that due to genetic
clustering.

## Shape and diet

PCA run on the frequency of occurrence of eight prey categories along nine sample localities
showed a clear distinction among the diet of otters living in coastal, islands, or mainland fresh-
water habitats, with the first axis explaining the 97.74% of cumulative variance (S1 Fig). The
diet of otters living along the mainland coasts differed from that of otters living on islands or
in mainland freshwaters. PC1 axis was mainly influenced by the relative frequency of freshwa-
ter (negative extreme) and marine fish (positive extreme), whereas PC2 axis was positively
associated with the relative frequency of crustaceans (S1 Fig). Otters living in coastal areas and

**Table 1. Association of A. Skull size (lnCS) and B. Shape, with genetic cluster, sex, and their interaction. Results are based on ANOVA model for size, and Procrustes ANOVA for shape.**

| A. lnCS (ANOVA) | Df | SS | MS | Rsq | F | Z | Pr(>F) |
|---|---|---|---|---|---|---|---|
| Genetic cluster | 2 | 0.019 | 0.009 | 0.083 | 2.228 | 1.235 | 0.107 |
| Sex | 1 | 0.049 | 0.049 | 0.217 | 11.630 | 2.660 | 0.003 |
| Genetic cluster:Sex | 2 | 0.002 | 0.001 | 0.008 | 0.223 | -0.859 | 0.799 |
| Residuals | 37 | 0.157 | 0.004 | 0.691 | | | |
| Total | 42 | 0.227 | | | | | |
| **B. Shape (Procrustes ANOVA)** | | | | | | | |
| Genetic cluster | 2 | 0.010 | 0.005 | 0.164 | 4.054 | 5.824 | 0.001 |
| Sex | 1 | 0.002 | 0.002 | 0.031 | 1.553 | 1.573 | 0.069 |
| Genetic cluster:Sex | 2 | 0.003 | 0.002 | 0.058 | 1.430 | 1.672 | 0.053 |
| Residuals | 37 | 0.045 | 0.001 | 0.747 | | | |
| Total | 42 | 0.060 | | | | | |

islands fed predominantly on marine fish, those along coasts also on crustaceans, whereas otters living in mainland freshwaters fed primarily on freshwater fish.

PLS regression revealed a significant association between shape and diet (PLS, based on a matrix of 37 prey (i.e., 31 fish species plus 6 non-fish taxa): r-PLS = 0.86, p = 0.03), with otters

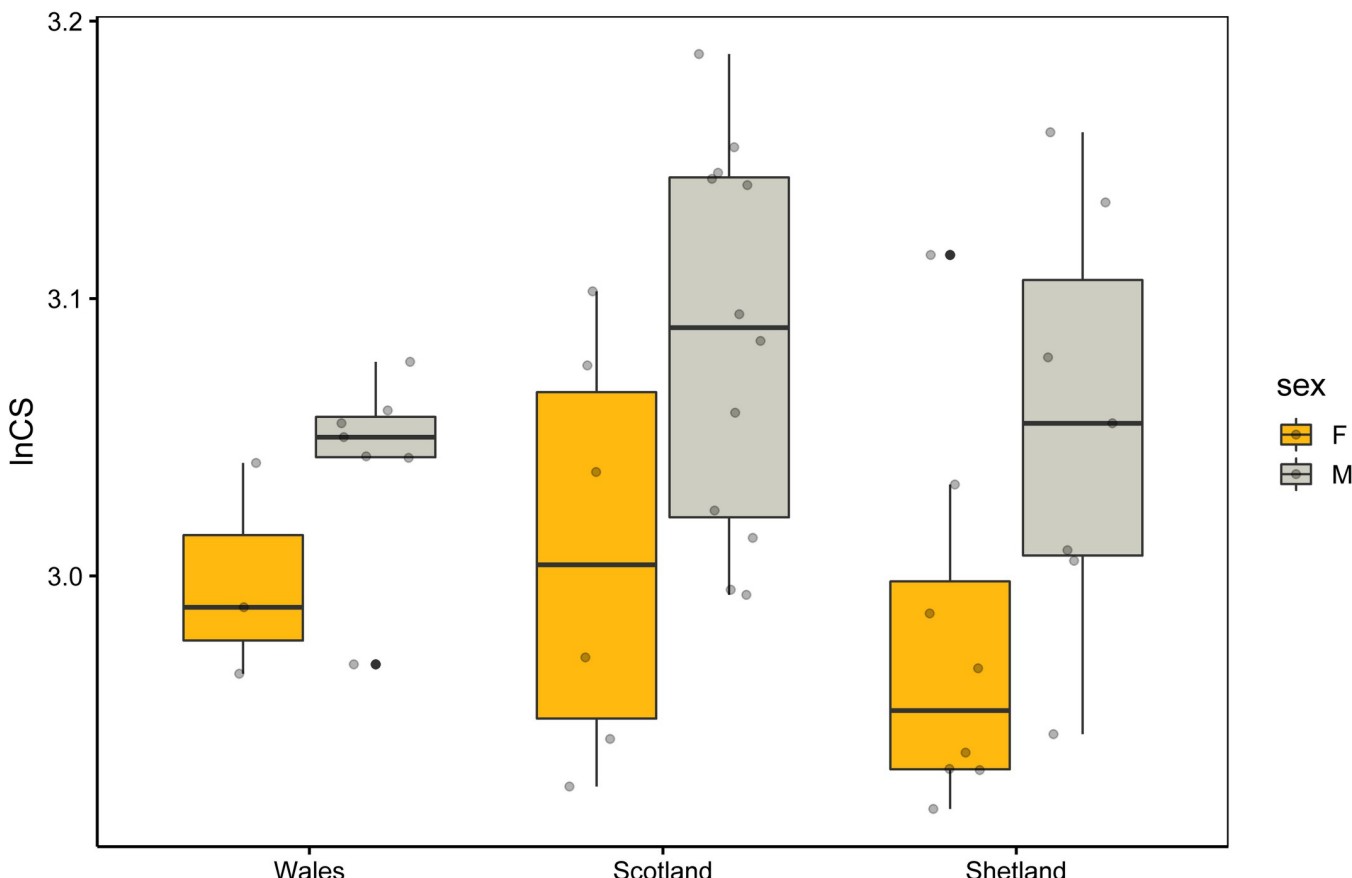

**Fig 3. Box plots of cranial size variation (= lnCS) in males and females from three genetic clusters.** Horizontal lines within each box indicate the median, upper and lower limits the inter-quartile range, while 'whiskers' indicate the minimum and the maximum.

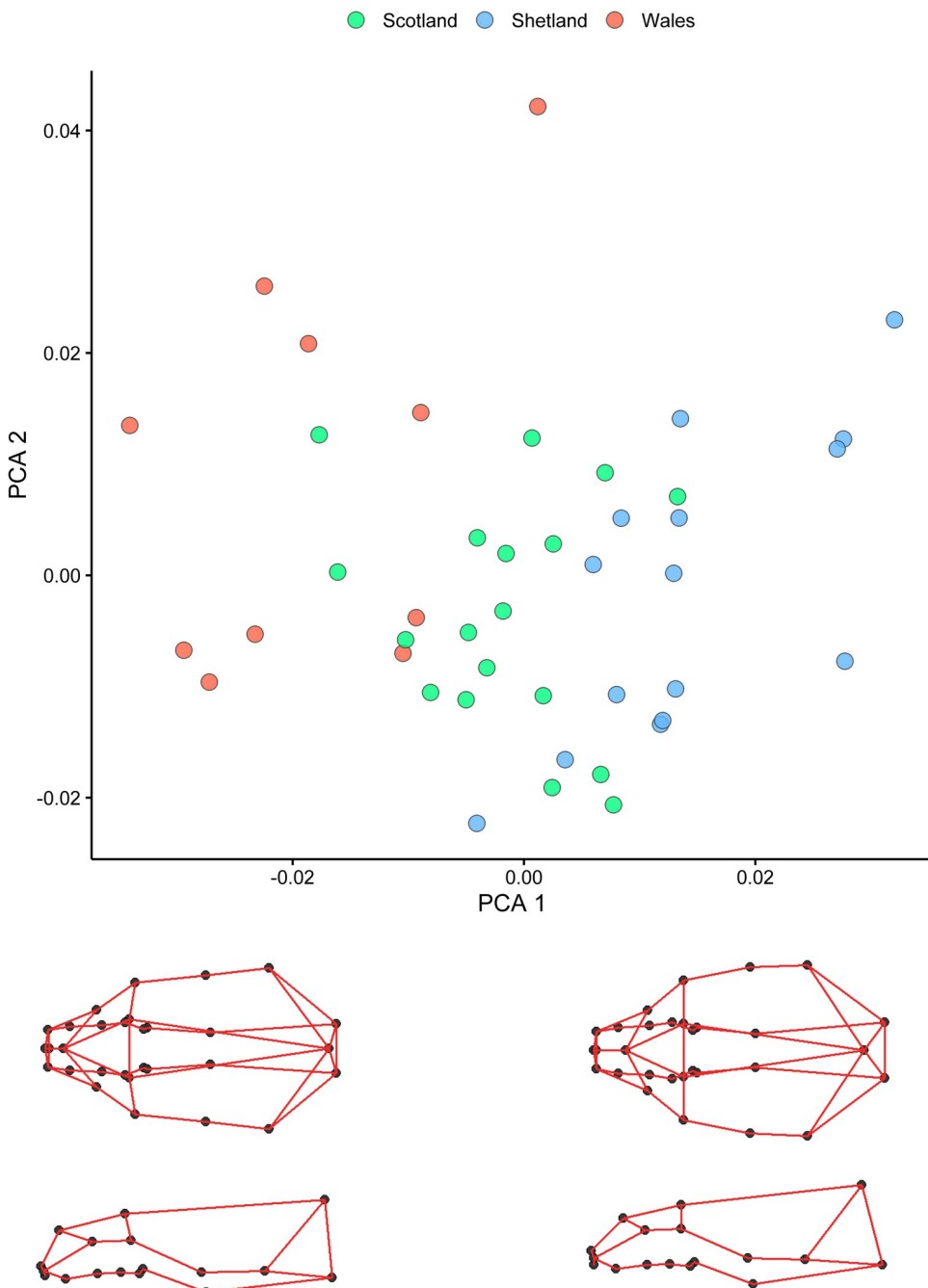

**Fig 4. Skull shape variation along the first two Principal Component axes (PCs) of Procrustes coordinates.** A. Colours indicate different genetic clusters, Scotland (green), Shetland (blue) and Wales (red). Individual data points represent the first two component scores from PCA carried out on all shape variables. B. A visualisation of the 3D contours related to extremes of variation along PC1, which describes 16.88% of the variance.

feeding in mainland freshwaters positioned at the positive extreme of both PLS1 and PLS2 axes, whereas marine fish feeders were found at the negative extreme of both axes, with the only exception of the sample from North East Scotland (Grampian). Samples from coastal habitats were either slightly distinguished or overlapped with the marine fish feeders from islands (Fig 5). No significant association found between cranial size and diet (r-PLS = 0.32, p = 0.83).

**Table 2. Association of skull shape with size (lnCS), genetic cluster and their interaction.** All the results are based on Procustes ANOVA models.

|  | Df | SS | MS | Rsq | F | Z | Pr(>F) |
|---|---|---|---|---|---|---|---|
| lnCS | 2 | 0.019 | 0.009 | 0.083 | 2.228 | 1.235 | 0.107 |
| Genetic cluster | 1 | 0.049 | 0.049 | 0.217 | 11.630 | 2.660 | 0.003 |
| lnCS: Genetic cluster | 2 | 0.002 | 0.001 | 0.008 | 0.223 | -0.859 | 0.799 |
| Residuals | 37 | 0.157 | 0.004 | 0.691 |  |  |  |
| Total | 42 | 0.227 |  |  |  |  |  |

The skulls of freshwater feeders were relatively slender and short-snouted compared to the skulls of otters feeding on marine fish, whereas the coastal otters, feeding on marine fish and crustaceans, showed an intermediate shape. Coastal and island otters also had larger orbits and eyes more oriented toward the bottom (LMs 3–5, 9–11 in Fig 2), a larger nasal cavity, and a larger distance between postorbital processes and zygomatic arch (Fig 6).

### Shape and climate

The variance inflation factors (VIF) identified significant correlation among most of the 19 bioclimatic variables. To avoid violating the assumption of independence, six unrelated variables were retained for further analyses: Annual Mean Temperature (BIO1), Isothermality (BIO3), Temperature Seasonality (BIO4), Mean Temperature of Wettest Quarter (BIO8), Mean Temperature of Driest Quarter (BIO9), Precipitation of Driest Month (BIO14) and Precipitation Seasonality (BIO15). PCA run on the six variables showed clear climate differences between the genetic clusters along the first two PC axes, expressing 99.11% of cumulative variation, and especially along PC1 (88.32% of cumulative variation) (S2 Fig). PC1 was mainly influenced by Temperature Seasonality (BIO4), whereas Precipitation of Driest Month (BIO14) and Precipitation Seasonality (BIO15) mainly affected variation along PC2 (S2 Fig). The Wales localities are clearly distinguished from Scotland and Shetland, being characterized by high temperature seasonality (BIO4). The remaining Scottish localities showed the maximum values. Moreover, samples from Shetland and one Scottish locality were characterized by low rainfall in the driest month.

Results from PLS analyses showed no correlation between cranial size and climate (r-PLS = -0.4966, P = 0.684), whereas climate was significantly related with the shape of the skull (r-PLS = 0.86, P = 0.003). Specifically, otters living in areas with a highest seasonality of temperature and precipitations had a more slender and long-snouted skull compared to otters living areas with low seasonality (Fig 7).

Shape changes from freshwater to marine fish feeding habits were similar to those observed in relation to climate, and along the geographic gradient. Comparison of PLS results indicated that although a larger effect size is associated with climate (PLS Z = 2.64) than with diet (PLS Z = 1.76) the difference between the two was not statistically significant (p = 0.75).

## Discussion

### Sexual dimorphism

As observed in most Musteloidea [7, 31, 32, 63, 64], we found a significant Sexual Size Dimorphism (SSD) in otter skulls from Great Britain, with males larger than females. Differences in skull size are usually related to bite force [30]. In species with polygyny mating systems, like otters, larger males with greater bite force are likely able to better defend their territory and mate with more females [65–68]. Another known advantage of SSD is the differentiation of trophic niches between males and females, lowering the resource competition between sexes

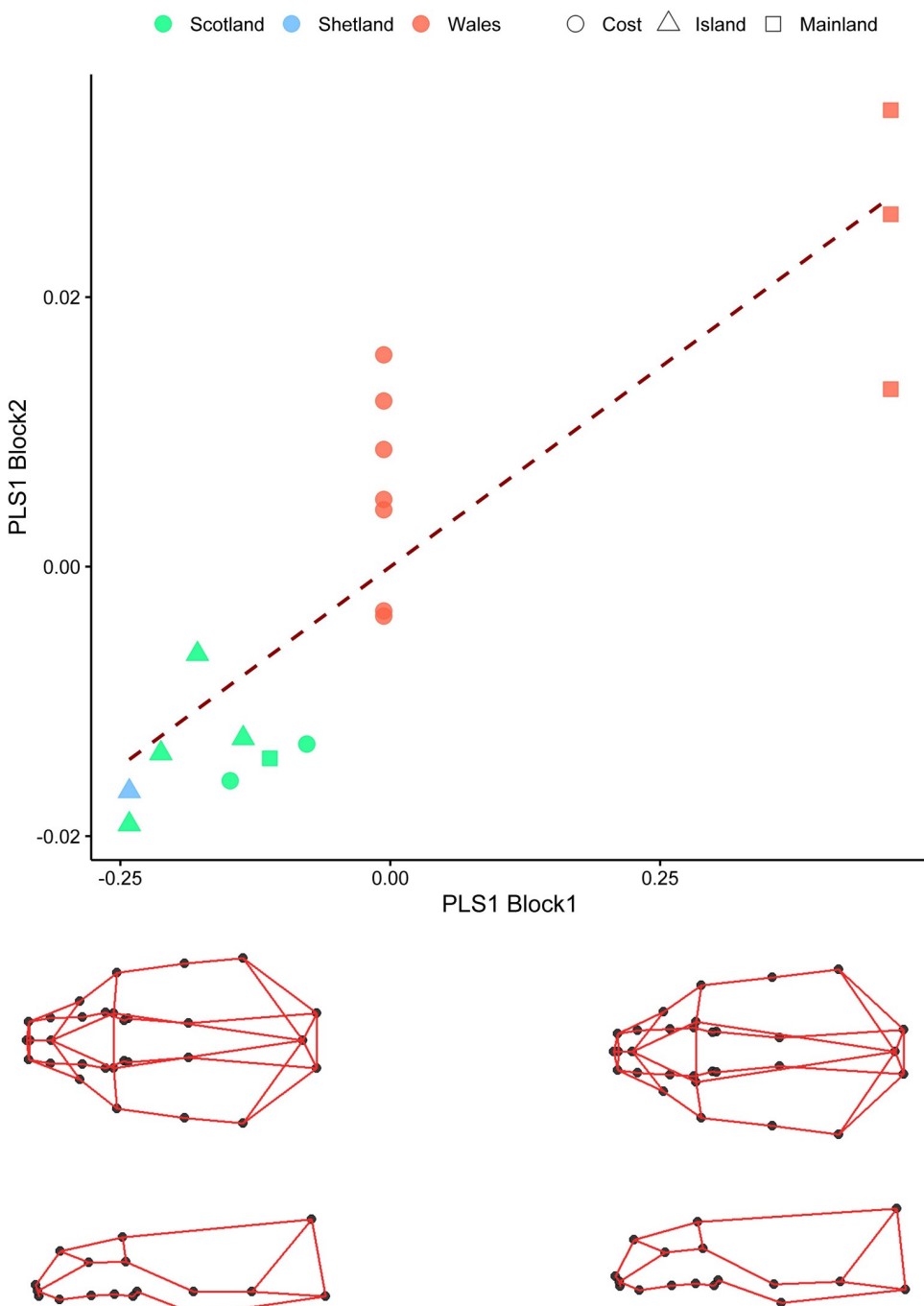

**Fig 5. Relationships between mean shape and diet as shown by the PLS regression of mean skull shape on diet matrices for 18 geographically distinct sampling locations.** Colours indicate the feeding areas of otters living in that locality. Below the x-axis are shown the wireframes corresponding to shape changes at the extremes of the axis. Nodes represent the landmarks, edges the anatomical connections between landmarks.

[19, 29]. In fact, it has been observed that both sexes and different age groups of otters feed on different types of prey or on prey of different sizes [26, 69]. Within this context, a larger size and a greater bite force could allow the males to hunt larger prey that are inaccessible to females. Although some previous studies using linear morphometry found a difference in the

**Wales**

**Shetland**

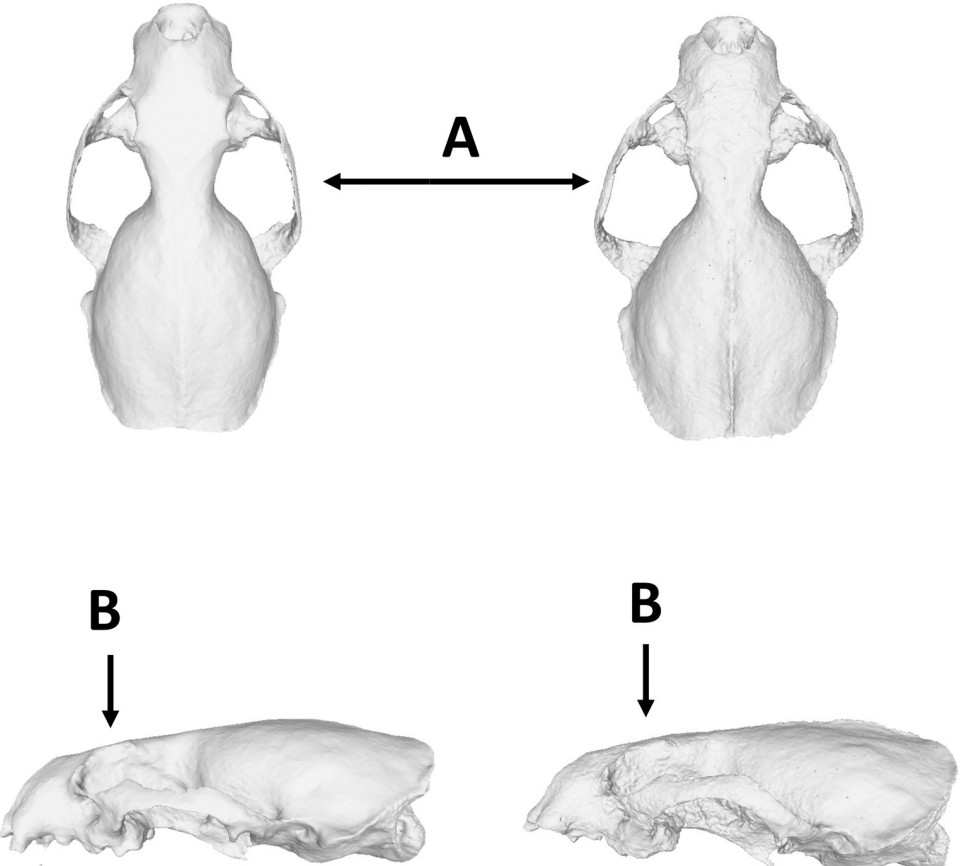

**Fig 6. Comparison of 3D models of a skull of an otter living in an island (right, Shetland, Coll ID: 1990.104.021) and an otter living in mainland freshwater (left, Wales, Coll ID: CARDIFF-733).** The arrow A indicates the zygomatic arch and B shows the postorbital processes.

shape of the skulls of males and females [11, 40, 70], we did not find any significant Sexual Shape Dimorphism (SShD). These contrasting results may reflect the different approaches used in the analyses, as GM is able to better capture the information on the shape compared to linear morphometry [71]. However, since shape differences between males and females were close to significance (p = 0.069), further analyses on a larger sample size might likely identify sexual related traits also in skull shape.

### Geographic variation

**Size.**  In contrast to the Bergmann's rule [72, 73] and body size variation of otters from Sweden [74], we did not find any latitudinal gradient in the skull size of otters from Great

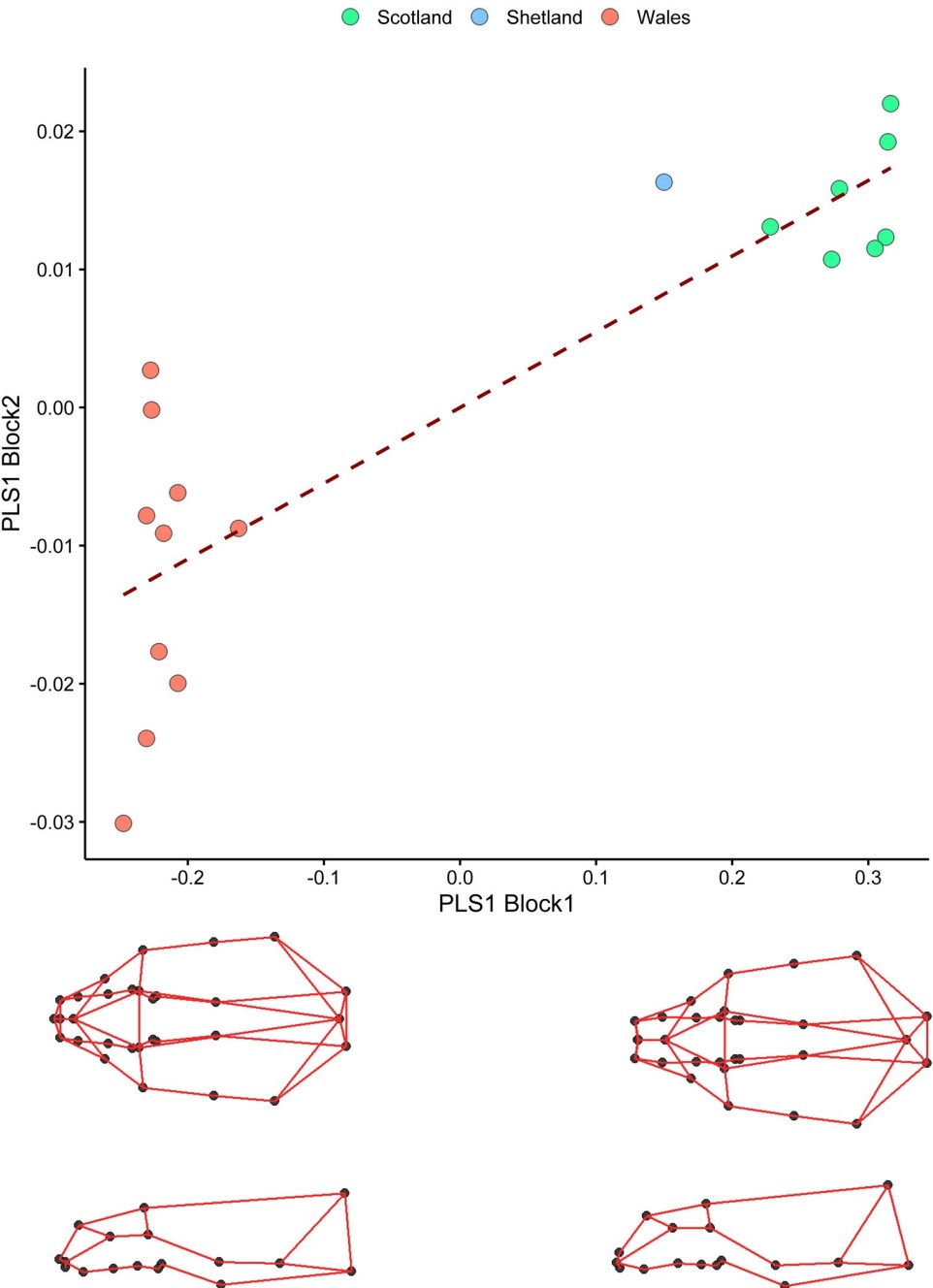

**Fig 7. Association between skull shape and climate.** Each point indicates one of the 18 sampling locations; colours indicate the genetic clusters. Data represent PLS scores from Block1 (representing mean shape variables for each sample locality) and Block2 (climate variables). Below the x-axis are shown the wireframes corresponding to shape changes at the extremes of the axis. Nodes represent the landmarks, edges the anatomical connections between landmarks.

Britain. Exceptions to Bergmann's rule have been found in many mammals [72, 75] and our sample now adds to previous evidences. The Shetland specimens could be subject to the island rule (i.e. smallest size [38]) and bias the gradient in size predicted by Bergmann's rule. Indeed, the lnCS of Scotland population appears to be larger on average than Wales.

**Shape.** Shape variation revealed a clear distinction between the three genetically distinct clusters (i.e., Shetland, Scotland and Wales), which are areas that support differences also in diet and climate.

We propose that the most likely driver of changes in skull shape are differences associated with the available diet. Our results evidenced that otters feeding on freshwater fish have a slender and short-snouted skull compared to marine fish feeders, whereas the increase of crustaceans in the diet of coastal otters is reflected in less marked 'marine' shape. A slender skull shape in freshwater feeding otters suggests a wider gape and a faster closure of the jaws [76]. Nevertheless, a more elongated braincase allows attachment for posterior and anterior temporalis muscles, increasing the horizontal force [77]. These traits are functional to capture fast swimming and soft prey like freshwater fish, especially salmonids. In contrast, otters feeding mainly in the marine environment have a sturdy skull with wider zygomatic arches. The wider zygomatic arches and taller crania in marine feeders can allow an increased area for the attachment of the masseter and temporalis jaw adductor muscles [30, 78]. These muscles function primarily to close the jaw [77] and the resulting larger temporalis mass in marine feeders can allow a stronger bite force [78, 22]. These characteristics are often attributed to durophagous otter species like the sea otter *Enhydra lutris* [76]. The diet of otters hunting in marine environments is mainly composed of benthic and slower swimming species (e.g., Zoarcidae and Lotidae (see S3 Table), with tougher skins, and reaching larger sizes compared to freshwater fish.

Marine and coastal feeding otters also differed in their larger orbits, larger nasal cavity and more bottom oriented eyes, which might be associated with a distinct hunting strategy compared to the inland feeders [79]. It seems plausible that the diurnal habit of the Shetland population [80] means that sight may be more important than in areas where nocturnal feeding is more typical and where water is characterized by higher turbidity. The compact shape of the skull and the larger orbits and large nasal cavity of marine feeding otters from Shetland may also represent an adaptation to favour the duration and depth of diving, as observed in pinnipeds [81]. Otters from Shetland have been observed diving to depths of more than 15m [82, 83]. In contrast, otters living in European freshwaters are mainly nocturnal [84–86], and are known to use their whiskers for hunting, as sight is impeded by both murky waters and darkness [87]. Also, freshwaters are not very deep but the currents can be rapid, and a slender skull may improve hydrodynamics and favour a higher swimming speed needed to catch fast swimming fish.

Skull traits associated with diet variation were similar to those associated with climate variation, as confirmed by comparison of PLS results. Climate is commonly used as a proxy of diet when this information is not available (see [52, 62, 88]) and our evidence suggests that climate may be used as a proxy for diet adaptation also in the Eurasian otter. On the other hand, Tseng and Flynn [79] indicated that skull shape variation in carnivores is correlated with precipitation, as this latter drives modifications in the sensory systems. In our case, the larger nasal cavity observed in otters living in the coldest areas, could increase oxygen assumption. This association is supported by evidence from Yom-Tov et al. [89] which shows negative association between water temperature and oxygen consumption in Eurasian otters.

**Concluding remarks.** Our study has highlighted how 3D morphometrics of the skull of otters across Great Britain mainland and islands was able to clearly differentiate the morphological characteristics of three distinct genetic clusters identified by Hobbs et al. [14] and Stanton et al. [13], and to identify the functional traits and possible drivers involved in the morphological shift within those areas. Our results suggest that the morphology of the otter skull can respond to adaptive pressures, that are likely related to the availability and accessibility of prey resources. That is, otters living in mainland freshwater, coastal areas and islands showed a clear distinction in the morphology of the skull, suggesting adaptive plasticity in

response to feeding resources. These findings are in contrast to the general belief that European otters are characterized by high homogeneity in their genetic and morphological traits, and highlight the need for more extensive investigations to identify any Evolutionary Significant Unit in need of conservation efforts to preserve the evolutionary potential in the species [90]. This issue is particularly relevant for a species like the Eurasian otter that lost most of its European populations in the past decades due to multiple anthropogenic pressures [2].

## Supporting information

**S1 File. Thirty 3D landmarks.**
(XLSX)

**S1 Fig. Spatial variation in the UK otters' diet.** Points represent 9 sampling locations for which diet composition was available, colours indicate genetic clusters. The first two axes of a Principal Component Analysis for percentage of seven prey categories (marine fish, freshwater fish, crustaceans, amphibians, reptiles, birds, mammals, and insects) are used to summarise variation between points.
(DOCX)

**S2 Fig. Spatial variation in climate between the three genetic clusters.** Each point is a location along Principal Component scores (PC1 vs PC2) from a PCA on the six bioclimatic variables overlapped in a bi-plot. PC1 and PC2 are primarily influenced by Temperature Seasonality (BIO4) and Precipitation of Driest Month (BIO 14), respectively.
(DOCX)

**S1 Table. Number of skull specimens analysed.** The sample is partitioned by sex for each geographic location.
(DOCX)

**S2 Table. Specimen list of the analysed otter skulls.** Abbreviations: NMS = National Museums of Scotland, M = male, F = female, A = Adult.
(DOCX)

**S3 Table. Percentage of dietary items from 9 sample localities.** Dietary data are based on.
(DOCX)

**S4 Table. Definition of climatic variables used in the analyses.** Abbreviations: BIO = bioclimate.
(DOCX)

## Acknowledgments

We are particularly grateful to Andrew Kitchener, principal curator of vertebrate biology at the National Museums Scotland who provided assistance and access to the otter collection.

## Author Contributions

**Conceptualization:** Luca Francesco Russo, Carlo Meloro, Elizabeth A. Chadwick, Anna Loy.

**Data curation:** Luca Francesco Russo, Mara De Silvestri, Anna Loy.

**Formal analysis:** Luca Francesco Russo, Carlo Meloro, Mara De Silvestri.

**Investigation:** Luca Francesco Russo, Carlo Meloro, Mara De Silvestri.

**Methodology:** Luca Francesco Russo, Carlo Meloro, Mara De Silvestri, Elizabeth A. Chadwick, Anna Loy.

**Project administration:** Carlo Meloro, Anna Loy.

**Software:** Luca Francesco Russo.

**Supervision:** Carlo Meloro, Anna Loy.

**Validation:** Carlo Meloro.

**Writing – original draft:** Luca Francesco Russo, Carlo Meloro.

**Writing – review & editing:** Luca Francesco Russo, Carlo Meloro, Elizabeth A. Chadwick, Anna Loy.

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
