## [Decision Letter · Decision Letter 0]

4 Jul 2022

PONE-D-22-15010Better sturdy or slender? Eurasian otter skull plasticity in response to feeding ecology and climatePLOS ONE

Dear Dr. Meloro,

Thank you for submitting your manuscript to PLOS ONE. After careful consideration, we feel that it has merit but does not fully meet PLOS ONE’s publication criteria as it currently stands. Therefore, we invite you to submit a revised version of the manuscript that addresses the points raised during the review process.

We look forward to receiving your revised manuscript.

Kind regards,

Bogdan Cristescu

Academic Editor

PLOS ONE

Academic Editor's (Bogdan Cristescu) Comments:

Two expert reviewers have provided positive assessment of the manuscript. Please address their comments in the revision.

The Abstract is a bit long, please provide less detail especially with regard to methodology.

Please consider a different color scheme for Fig. 3, as right now the colors coincide with some of the geographic localities from the other figures.

For clarity of Figs. 5 and 7, please indicate what the skull schematics (edges and nodes) as well as view angles represent to justify their inclusion.

Journal Requirements:

We are particularly grateful to Andrew Kitchener, curator of the Scottish Natural Museum who provided assistance and access to the otter collection. LFR was funded by PhD scholarship from University of Molise.  

NO authors have competing interests.

5. We note that Figure 1 in your submission contain map/satellite images which may be copyrighted. All PLOS content is published under the Creative Commons Attribution License (CC BY 4.0), which means that the manuscript, images, and Supporting Information files will be freely available online, and any third party is permitted to access, download, copy, distribute, and use these materials in any way, even commercially, with proper attribution. For these reasons, we cannot publish previously copyrighted maps or satellite images created using proprietary data, such as Google software (Google Maps, Street View, and Earth). For more information, see our copyright guidelines: http://journals.plos.org/plosone/s/licenses-and-copyright.

a) You may seek permission from the original copyright holder of Figure 1 to publish the content specifically under the CC BY 4.0 license.  

Natural Earth (public domain): http://www.naturalearthdata.com/.

Reviewers' comments:

Reviewer's Responses to Questions

**Comments to the Author**

1. Is the manuscript technically sound, and do the data support the conclusions?

Reviewer #1: Yes

Reviewer #2: Yes

2. Has the statistical analysis been performed appropriately and rigorously? 

Reviewer #1: Yes

Reviewer #2: Yes

3. Have the authors made all data underlying the findings in their manuscript fully available?

Reviewer #1: Yes

Reviewer #2: Yes

4. Is the manuscript presented in an intelligible fashion and written in standard English?

Reviewer #1: Yes

Reviewer #2: Yes

5. Review Comments to the Author

Reviewer #1: The authors did a great job with the methodology and results section. I had very few minor comments and two major comments for the discussion.

Minor Comments

Line 46: Add a comma between “mammals and skull shape” or change to “Several studies have shown that the skull shape in mammals is closely related…”

Line 51: Add a comma between “expected” and “especially”.

Line 62: Maybe removed the sentence “Here, we used 3D GM…” as this is stated again in the next paragraph when describing the objectives of the research. It sounded redundant.

Line 96: What is the fixed distance on the tripod?

Line 103: Add a space between “data” and “[41,47]”.

Line 114: Add a period after “[48,49]”

Line 125: Add a comma between “size or shape” and “we used…”

Line 140: Add space between “factors” and “[56]”

Line 141: Be consistent with subtitles. I recommend using “Shape and climate.” to be consistent with the “Shape and diet.” in line 128.

Line 156: Add “being” between “always” and “larger”… “with male skulls always being larger than female’s…”

Line 224: Change “an” to “and” and put a comma after “Shetland”?

Line 225: Remove “,” after (BIO4). Capitalize “the”. I believe this is a new sentence.

Line 225: Remove the space between “values” and the period.

Line 226: “scottiesh”?

Line 248: Add a space between “sizes” and “[25,67]”

Line 253: Add space between “morphometry” and “[69]”

Line 269: Change “durophage” to “durophagous”.

Line 291: Name the authors for reference 14, like Stanton et al. 2014 to be consistent.

Line 294: Remove space between “resources” and the period.

Major Comments: I would like to see more information in the discussion section.

Line 51-53: The authors mention that cranial and mandibular morphology are expected to have modifications in dentition and masticatory muscle attachment area. It would be a nice addition to include hypotheses in how these morphological modifications affect the masticatory muscles in the discussion based on diet. A lot can be discussed on muscle attachment and the feeding biomechanics per group based on dietary differences.

The authors showed that there was variation among skull morphology based on climate, but it is not well discussed in the discussion area. It would be a great addition to the overall manuscript if this was better explained in the discussion with more detail. I was interested in knowing more details about climate variation among the groups and thought it would be better explained in the discussion. How did climate specifically affect the morphology between the groups?

Reviewer #2: Manuscript review

Better sturdy or slender? Eurasian otter skull plasticity in response to feeding ecology and climate

Line 49: Add cite to the sentence.

Line 96: Change the citation (Loy et al. 2021) to the journal numeric format.

Line 131: Change (Suppl. Mat. 4) for (Suppl. Mat. 3).

Line 141. Change (Suppl. Mat. 3) for (Suppl. Mat. 4).

Line 189. Change (Suppl. Mat. 4) for (Suppl. Mat. 5).

Line 223. Change (Suppl. Mat. 5) for (Suppl. Mat. 6).

Introduction:

This sentence is a bit repetitive with the paragraph in line 71: “Here, we used 3D GM of the skull to investigate the morphological variation of otters across the mainland and the islands of Great Britain, and to explore the ultimate drivers of the observed patterns in terms of the genetic, latitudinal and ecological differentiation revealed by recent studies”. Change the wording so that the information does not sound repetitive.

Supplementary material:

The legend of Supp. Mat. 6 need check the redaction and format.

Questions

All your specimens were adult individuals or did you not consider that variable? since it can affect the results, especially in the variation in size between males and females.

It is true that they did not find a relationship with Bergman's rule, but they did not consider that the specimens that lives in Shetland are on an island far from the mainland and that several species of mammals that inhabit islands have dwarfism or smaller sizes than their congeners. who live on mainland.

I suggest that you further enrich the discussion of paragraph 273, I think you could cite works related to climatic-diet-morphology variation to further support your discussion.

6. PLOS authors have the option to publish the peer review history of their article (what does this mean?). If published, this will include your full peer review and any attached files.

Reviewer #1: No

Reviewer #2: No

---

## [Author Response · Author response to Decision Letter 0]

2 Aug 2022

We have followed all the suggestions provided by both reviewers and we appreciate all their constructive criticisms. We have improved the Abstract section and the manuscript in general following your advise and at the bottom of this letter you will find our answers (preceded by the suffix: ANSWER) to each of the raising points of criticisms that were mostly stylistic and conceptual. 

We note that Figure 1 in your submission contain map/satellite images….

ANSWER: The map was obtained from www.data.gov.uk under the Open Government Licence v3.0 (https://www.nationalarchives.gov.uk/doc/open-government-licence/version/3/). We added this paragraph in the legend of Fig 1:

Country Boundaries was downloaded from www.data.gov.uk under the Open Government Licence v3.0

Academic Editor's (Bogdan Cristescu) Comments:

Two expert reviewers have provided positive assessment of the manuscript. Please address their comments in the revision.

ANSWER: Dear Editor, we are truly grateful for your suggestions that helped to greatly implement the manuscript and to clarify the contents of our study. Here following are detailed answers to the comments by the editor and the reviewers.

The Abstract is a bit long, please provide less detail especially with regard to methodology.

ANSWER: We modified the abstract according to your request. 

Please consider a different color scheme for Fig. 3, as right now the colors coincide with some of the geographic localities from the other figures.

ANSWER: Done

For clarity of Figs. 5 and 7, please indicate what the skull schematics (edges and nodes) as well as view angles represent to justify their inclusion.

ANSWER: We added the following in the legends of fig 5 and 7:Below the x-axis are shown the wireframes corresponding to shape changes at the extremes of the axis. Nodes represent the landmarks, edges the anatomical connections between landmarks.

Reviewer #1: The authors did a great job with the methodology and results section. I had very few minor comments and two major comments for the discussion.

ANSWER: Dear Reviewer, thank you for your comments which have helped us improve our work

Minor Comments

Line 46: Add a comma between “mammals and skull shape” or change to “Several studies have shown that the skull shape in mammals is closely related…”

ANSWER: Done

Line 51: Add a comma between “expected” and “especially”.

ANSWER: Done

Line 62: Maybe removed the sentence “Here, we used 3D GM…” as this is stated again in the next paragraph when describing the objectives of the research. It sounded redundant.

ANSWER: Done

Line 96: What is the fixed distance on the tripod?

ANSWER: We modified the sentence: placed on the tripod at a fixed distance (50 cm) from the turntable

Line 103: Add a space between “data” and “[41,47]”.

ANSWER: Done

Line 114: Add a period after “[48,49]”

ANSWER: Done

Line 125: Add a comma between “size or shape” and “we used…”

ANSWER: Done

Line 140: Add space between “factors” and “[56]”

ANSWER: Done

Line 141: Be consistent with subtitles. I recommend using “Shape and climate.” to be consistent with the “Shape and diet.” in line 128.

ANSWER: Done

Line 156: Add “being” between “always” and “larger”… “with male skulls always being larger than female’s…”

ANSWER: Done

Line 224: Change “an” to “and” and put a comma after “Shetland”?

ANSWER: Done

Line 225: Remove “,” after (BIO4). Capitalize “the”. I believe this is a new sentence.

ANSWER: Done

Line 225: Remove the space between “values” and the period.

ANSWER: Done

Line 226: “scottiesh”?

ANSWER: We replace “scottiesh” with “Scottish”

Line 248: Add a space between “sizes” and “[25,67]”

ANSWER: Done

Line 253: Add space between “morphometry” and “[69]”

ANSWER: Done

Line 269: Change “durophage” to “durophagous”.

ANSWER: Done

Line 291: Name the authors for reference 14, like Stanton et al. 2014 to be consistent.

ANSWER: Done

Line 294: Remove space between “resources” and the period.

ANSWER: Done

Major Comments: I would like to see more information in the discussion section.

Line 51-53: The authors mention that cranial and mandibular morphology are expected to have modifications in dentition and masticatory muscle attachment area. It would be a nice addition to include hypotheses in how these morphological modifications affect the masticatory muscles in the discussion based on diet. A lot can be discussed on muscle attachment and the feeding biomechanics per group based on dietary differences.

ANSWER: Thanks for your suggestion, we added this paragraph at line 286: Nevertheless, a more elongated braincase allows attachment for posterior and anterior temporalis muscles, increasing the horizontal force [77] ; and this paragraph at line 289 : The wider zygomatic arches and taller crania in marine feeders can allow an increased area for the attachment of the masseter and temporalis jaw adductor muscles [30,78]. These muscles function primarily to close the jaw [77] and the resulting larger temporalis mass in marine feeders can allow a stronger bite force [78,79]

The authors showed that there was variation among skull morphology based on climate, but it is not well discussed in the discussion area. It would be a great addition to the overall manuscript if this was better explained in the discussion with more detail. I was interested in knowing more details about climate variation among the groups and thought it would be better explained in the discussion. How did climate specifically affect the morphology between the groups?

ANSWER: We added the following paragraph to the discussion in line 311. Skull traits associated with diet variation were similar to the traits associated with climate variation, as confirmed by comparison of PLS results. Climate is commonly used as a proxy of diet when this information is not available (see [52,62,89]) and our evidence suggests that climate may be used as a proxy for diet adaptation in the Eurasian otter also. On the other hand, Tseng and Flynn [90] indicated that skull shape variation in carnivores is correlated with precipitation, as this latter drives modifications in the sensory systems. In our case, the larger nasal cavity observed in otters living in the coldest areas, could increase oxygen assumption. This association is supported by evidence from Yom-Tov et al. [91] which shows negative association between water temperature and oxygen consumption in Eurasian otters

Reviewer #2: Manuscript review

Better sturdy or slender? Eurasian otter skull plasticity in response to feeding ecology and climate

Line 49: Add cite to the sentence.

ANSWER: Done

Line 96: Change the citation (Loy et al. 2021) to the journal numeric format.

ANSWER: Done

Line 131: Change (Suppl. Mat. 4) for (Suppl. Mat. 3).

ANSWER: Done

Line 141. Change (Suppl. Mat. 3) for (Suppl. Mat. 4).

ANSWER: Done

Line 189. Change (Suppl. Mat. 4) for (Suppl. Mat. 5).

ANSWER: Done

Line 223. Change (Suppl. Mat. 5) for (Suppl. Mat. 6).

ANSWER: Done

Introduction:

This sentence is a bit repetitive with the paragraph in line 71: “Here, we used 3D GM of the skull to investigate the morphological variation of otters across the mainland and the islands of Great Britain, and to explore the ultimate drivers of the observed patterns in terms of the genetic, latitudinal and ecological differentiation revealed by recent studies”. Change the wording so that the information does not sound repetitive.

ANSWER: Done 

Supplementary material:

The legend of Supp. Mat. 6 need check the redaction and format.

Questions

All your specimens were adult individuals or did you not consider that variable? since it can affect the results, especially in the variation in size between males and females.

ANSWER: Yes they are all adults, as reported at line 89 in the methods 

It is true that they did not find a relationship with Bergman's rule, but they did not consider that the specimens that lives in Shetland are on an island far from the mainland and that several species of mammals that inhabit islands have dwarfism or smaller sizes than their congeners. who live on mainland. 

ANSWER: Thanks for your comment, we added this paragraph in the line 276 : [72,75]. The Shetland specimens could be subject to the island rule (i.e. smallest size [38]) and bias the gradient in size predicted by Bergmann rules, in fact, the lnCS of Scotland population seems to be larger than Wales. 

I suggest that you further enrich the discussion of paragraph 273, I think you could cite works related to climatic-diet-morphology variation to further support your discussion.

ANSWER: We added the following paragraph to the discussion in line 311. Skull traits associated with diet variation were similar to the traits associated with climate variation, as confirmed by comparison of PLS results. Climate is commonly used as a proxy of diet when this information is not available (see [52,62,89]) and our evidence suggests that climate may be used as a proxy for diet adaptation in the Eurasian otter also. On the other hand, Tseng and Flynn [90] indicated that skull shape variation in carnivores is correlated with precipitation, as this latter drives modifications in the sensory systems. In our case, the larger nasal cavity observed in otters living in the coldest areas, could increase oxygen assumption. This association is supported by evidence from Yom-Tov et al. [91] which shows negative association between water temperature and oxygen consumption in Eurasian otters

---

## [Editor Report · Decision Letter 1]

7 Sep 2022

Better sturdy or slender? Eurasian otter skull plasticity in response to feeding ecology and climate

PONE-D-22-15010R1

Dear Dr. Meloro,

We’re pleased to inform you that your manuscript has been judged scientifically suitable for publication and will be formally accepted for publication once it meets all outstanding technical requirements.

Kind regards,

Bogdan Cristescu

Academic Editor

PLOS ONE

Additional Editor Comments (optional):

The authors adequately incorporated the suggestions from the reviewers and academic editor. Congratulations on your paper.
---

## [Editor Report · Acceptance letter]

20 Sep 2022

PONE-D-22-15010R1 

Better sturdy or slender? Eurasian otter skull plasticity in response to feeding ecology 

Dear Dr. Meloro:

I'm pleased to inform you that your manuscript has been deemed suitable for publication in PLOS ONE. Congratulations! Your manuscript is now with our production department. 

Kind regards, 

on behalf of

Dr. Bogdan Cristescu 

Academic Editor

PLOS ONE